# No Influence of Mechatronic Poles on the Movement Pattern of Professional Nordic Walkers

**DOI:** 10.3390/ijerph20010163

**Published:** 2022-12-22

**Authors:** Agnieszka Szpala, Sławomir Winiarski, Małgorzata Kołodziej, Bogdan Pietraszewski, Ryszard Jasiński, Tadeusz Niebudek, Andrzej Lejczak, Karolina Lorek, Jacek Bałchanowski, Sławomir Wudarczyk, Marek Woźniewski

**Affiliations:** 1Department of Biomechanics, Wroclaw University of Health and Sport Sciences, Mickiewicza 58 Street, 51-684 Wrocław, Poland; 2Department of Human Biology, Wroclaw University of Health and Sport Sciences, Paderewskiego 35 Avenue, 51-612 Wrocław, Poland; 3Department of Physical Culture Pedagogy, Wroclaw University of Health and Sport Sciences, Paderewskiego 35 Avenue, 51-612 Wrocław, Poland; 4Department of Physiotherapy in Surgical Medicine and Oncology, Wroclaw University of Health and Sport Sciences, Paderewskiego 35 Avenue, 51-612 Wrocław, Poland; 5Department of Kinesiology, Wroclaw University of Health and Sport Sciences, Paderewskiego 35 Avenue, 51-612 Wrocław, Poland; 6Department of Fundamentals of Machine Design and Mechatronics Systems, Wroclaw University of Science and Technology, Łukasiewicza 7/9 Street, 50-371 Wrocław, Poland

**Keywords:** gait analysis, movement pattern, human performance, Nordic Walking, mechatronic poles, biomedical signal

## Abstract

This study compared selected temporal and kinematic parameters of normal gait and Nordic Walking (NW) performed with classic and mechatronic poles (classic poles equipped with sensors). It was assumed that equipping NW poles with sensors for biomechanical gait analysis would not impair the NW walking technique. Six professional NW instructors and athletes, including three women, participated in the study. The MyoMotion MR3 motion analysis system was used to collect gait kinematic variables. The subject’s task was to cover a 100-m distance with three types of gait: a gait without poles, a gait with classic NW poles, and a gait with mechatronic poles at the preferred speed. Parameters were measured both on the right and left sides of the body. No significant differences were found between gait types for three temporal parameters: step cadence, step, and stride time. For the other variables, all the differences identified were between free-walking and walking with poles, with no differences between standard and mechatronic poles. For nine kinematic parameters, differences between free-walking and walking with poles for both the left and right sides were found, while no differences were due to the pole type. All temporal parameters were characterized by symmetry, while among kinematic parameters, only two were asymmetrical (shoulder abduction–adduction in walking with regular poles and elbow flexion–extension in walking without poles). Equipping classic NW poles with additional signaling and measuring devices (mechatronic poles) does not impair the NW technique, making it possible to use them in further studies of gait biomechanics.

## 1. Introduction

Since ski runners in Finland in the 1920s began using special poles during their walking training, Nordic Walking (NW) has become a popular activity in sports and recreation and rehabilitation [1,2]. The addition of the active use of a pair of poles during walking results in changes in walking distance, speed, and muscle activity, especially of the upper body, and gait kinematics [3]. Studies conducted so far show an increase in walking distance with poles by 10% and speed by 25% compared to walking without poles. They also found a 10% increase in stride length and a 6% decrease in cadence during NW gait, although other studies have found different results: a 13% reduction in stride length and a 14% increase in cadence. This ambiguity in the results confirms the need for research that has reliably developed a gait pattern with NW poles. Single- and double-support times also increased, by almost 10% and more than 20%, respectively, resulting in a 21% increase in total support time in NW gait [3,4,5]. NW increases activity primarily in the upper limbs and trunk muscles and, in functional strength tests, also in the lower limbs. The range of motion in the joints of the upper and lower extremities was also increased during NW gait, while the results for the trunk were inconclusive. Some authors found a reduction in the range of motion of the trunk in the sagittal plane and some in the frontal plane. Further, pelvic tilt and torsion increased during the NW gait compared to the non-pole gait [3,6,7,8,9]. NW also increases the reaction of ground forces in all axes except the vertical axis when pushing back with a pole [4]. In addition, NW improves gait economy and increases muscle engagement and coordination, being a safe form of physical training used in many rehabilitation specialities, including cardiology, oncology, geriatrics, and neurology [10].

Our studies on the use of NW in patients with intermittent claudication show that the effectiveness of this training depends mainly on proper gait technique, which can significantly impact rehabilitation outcomes [11,12]. When using NW, attention is mainly paid to physiological parameters, including distance, gait speed, and intensity, without considering its kinematic and kinetic parameters. This, moreover, is difficult using classic NW poles. Therefore, work is underway on the design of mechatronic poles equipped with sensors that would allow real-time analysis of these parameters and based on this, correction of the gait technique. Such a system could identify the basic parameters that characterize correct movement technique and provide feedback to the user of the poles. 

One such system component is a wireless communication module that transmits the measured NW gait parameters to a mobile device (smartphone/tablet). A dedicated application is installed on the mobile device, which enables online/offline monitoring and analysis of NW gait by the trainer, physiotherapist, or patient [13]. Among other things, the app enables the visualization of selected waveforms in the form of dynamic graphs and displays animations of pole movement. The application includes a module for defining events, the occurrence of which is signaled by sound/vibration (e.g., signaling the moment the pole contacts the ground or is released); it can define the angular range of the movement and signal its exceedance, etc.) [13]. There are isolated studies in the literature evaluating the usefulness of using sensors in natural conditions in motion analysis, but mainly in cross-country skiing. Despite some similarities in movement technique in cross-country skiing and NW, there are differences that call for developing such systems exclusively for walking with poles. The main considerations are the angle between the ground and the pole during the push-off phase, the duration of the movement cycle, and the reaction force of the ground during contact with the pole. The few research results to date support the usefulness of such a system in movement analysis during NW [14]. What is unknown is the effect of mechatronic poles on the movement technique and kinematic movement patterns of NW practitioners. 

Therefore, the purpose of this study was to evaluate selected biomechanical parameters of the gait of NW instructors and athletes as performed (1) without poles, (2) with classic NW poles, and (3) with mechatronic poles (classic poles equipped with electronic sensors). It was assumed that equipping the NW poles with additional electronic sensors would not significantly alter the NW walking technique (movement pattern), allowing them to be used in subsequent scientific studies. A system for the use of NW poles has not yet been developed to provide feedback on the basic kinematic and dynamic parameters of gait and to develop an optimal model of gait with poles and, consequently, a model of this gait for people with various dysfunctions. Effectiveness, safety, accessibility, low financial cost, the possibility of self-exercise, and the selection of intensity to individual capabilities are factors recommending this form of exercise in the primary and secondary prevention of many diseases.

## 2. Materials and Methods

### 2.1. Study Design

This study was an observational case series study with adopted convenience sampling. 

### 2.2. Participants

Six NW instructors and athletes, including three women, participated in the study. The characteristics of the participants are shown in Table 1.

All participants were instructors or coaches of the International Nordic Walking Federation (INWA) and were characterized by very long—for a young discipline such as NW—seniority in this discipline (an average of 13.5 years), outstanding sporting achievements (e.g., Case4, 4× Polish Championship; Case6, 2× Polish Championship; Case 2, 1× Polish Championship), outstanding achievements in assessing the correctness of gait technique during NW competitions (e.g., Case5, 10× during Polish Championships; Case3, 10× during Polish Championships; Case6, 5× during Polish Championships). The study group also included participants in many international training courses in the NW technique (Case6: Finland, the Netherlands, Germany, Slovenia; Case2: Ukraine, Russia, Estonia, Finland). The above data indicate clearly that those included in the research group of “professionals” represented the highest level of knowledge of correct NW technique. All study participants were healthy and without any injuries affecting their natural movement. They entered the study consecutively, and the measurement conditions were the same. 

The participants were informed about the aims and methodology used in the experiment and gave written informed consent for participation in the investigation. The experiment was approved by the local ethics committee and conducted in accordance with the Declaration of Helsinki.

### 2.3. Research Methods

The MyoMotion MR3 motion analysis system (Noraxon Inc., Scottsdale, AZ, USA) collected gait kinematic variables. It is a set for three-dimensional motion evaluation using inertial measurement unit (IMU) sensors. The system combines wireless data transmission and IMU sensor technology to evaluate any motion in 3-dimensional space (e.g., changes in angles between segments and linear acceleration). Each of the sensors included in the kit combines an accelerometer, a gyroscope, and a sensor for measuring the Earth’s magnetic field. The sensors are wireless, with a transmission range of up to 30 m. Presently, it is recognized that Noraxon’s IMUs can measure angles with a static accuracy of ±0.4 deg and dynamic accuracy of ±1.2 deg [15,16]. The IMU sensors were placed on the subject’s body according to a model compatible with MR3 software (Figure 1), which enabled both data recording and comprehensive data analysis (the sampling rate used in the study was 200 Hz). To collect gait data, a total of 16 sensors were attached: three sensors for each of the upper and lower extremities (right/left), three sensors in the spinal region (on the spinous process of the 7th cervical and 7th thoracic vertebrae and in the sacral region), and one sensor on the forehead (Figure 1). Calibration of the IMU sensor for body position was performed before each measurement. A standing position with arms parallel along the torso was used to determine the value of the 0° angle as a calibration posture.

The subject’s task was to walk a 100-m distance in three ways: (1) freely and without poles (normal walking), (2) with classic NW poles, and (3) with mechatronic poles. In each case, they started with a free gait, and the order was random for the walk with poles. The walk took place on a pitch with an artificial surface at what is known as natural (preferred) speed. Two 100-m passes were made for each gait type, allowing an average of 70 complete gait cycles (full strides) per pass. Short rest periods were used between trials to eliminate the effect of fatigue.

The design of mechatronic poles and their characteristics have been described in detail in our previous works [13,17]. In short, the classic NW poles were equipped with two 9-axis inertial sensors (3-axis gyroscope, accelerometer, and magnetometer). The pressure sensor measuring a force along the pole’s axis was mounted within the foot of the pole, and a contact sensor is mounted in the handle. Tests conducted on inertial, pressure, contact, and distance sensor signals confirmed sufficient accuracy for gait biomechanics studies [13,17,18].

To obtain more information on the human locomotion, additional IMU sensors were used to register the movements of human body parts. Following the International Society of Biomechanics recommendations [19,20,21], the following parameters were recorded separately for the right (RT) and the left (LT) side: Fifteen temporal parameters characterizing the structure of the step cycle: step cadence (in step/min), characterizing mean number of steps per minute (step rate); stride time duration (s), as the time elapsed between the initial contacts of two consecutive footfalls of the same foot and characterizing mean cycle time; step time duration (s), as the time elapsed from the initial contact of one foot to the initial contact of the opposite foot; stance and swing phase relative duration (%), as the stance time normalized to stride time; single-support relative duration (%), as the time elapsed when only one foot is in contact with the ground between the last contact of the opposite footfall to the initial contact of the next footfall of the same foot, normalized to stride time; double support (%), as the sum of times when both feet are in contact with the ground simultaneously (twice at initial and terminal stance); loading response (%) (also known as “foot flat”), as the double-support time measured from initial contact until the contralateral foot leaves the ground (contralateral toe off); pre-swing (%), as the double-support time from the contralateral foot contact until the ipsilateral foot leaves the ground.Mean Range of Motion (ROM) in whole cycle expressed in angular degrees (deg), characterizing movements of the upper and lower extremities relative to the global coordinate system. In particular,for the upper extremity: shoulder flexion–extension, anterior or posterior movement of the humerus relative to the thorax in the sagittal plane; shoulder abduction–adduction, movement of the humerus relative to the thorax in the frontal plane; shoulder internal–external rotation, rotation of the humerus in the transversal plane; elbow flexion–extension, movement of the forearm relative to the humerus along the transversal axis; wrist flexion–extension, movement of wrist relative to the radius along the transversal axis and measured between the upper arm and hand sensors; wrist radial–ulnar, movement of wrist relative to the radius and measured between the upper arm and hand sensors; wrist supination–pronation, movement of wrist relative to the radius along the axis and measured between the upper arm and hand sensorsfor the lower extremity: hip flexion–extension, movement of the femur in the sagittal plane about the mediolateral axis; hip ab/adduction, movement of the femur with respect to the pelvis in the frontal plane; hip rotation, movement of the femur in the transversal plane due to rotation around the proximal–distal axis; knee flexion–extension, movement of the tibia with respect to the femur in the sagittal plane; ankle dorsi–plantarflexion, movement of the foot with respect to the tibia in the sagittal plane; ankle ab/adduction, movement of the foot in the transverse (global) plane; ankle inversion–eversion, movement of the foot in the frontal (global) plane.

Gait cycles and phases were determined using what are known as virtual footswitches. Virtual footswitches allow the user to visualize a subject’s steps and have the added benefit of utilizing the dedicated Gait Foot Switch Report.

### 2.4. Statistical Analysis

The values of all measurements are presented in tables as mean ± standard deviation (M ± SD). The normality of the variables was checked with the Shapiro–Wilk test. Not all variables had a normal distribution, so we also used non-parametric methods, which are shown in italics in Table 2 and Table 3. Differences between parameters measured in 3 measurements (during the free walk and the walk with standard and mechatronic poles) were checked by analysis of variance for repeated measurements with the right and left side (limb) as a possible differentiating factor for the variables studied. Homogeneity of variance was verified with the M-Box test, sphericity was verified with the Mauchly test, and Greenhouse–Geisser correction and multivariate tests with Wilks correction were applied where necessary. Tukey’s post hoc test was used for multiple comparisons. The Friedman and Dunn–Bonferroni post hoc tests were used if the ANOVA assumptions were violated. 

All analyses were performed using Statistica 13.3.0 (TIBCO Software Inc., Palo Alto, CA, USA). The statistical significance of the results was accepted if *p* < 0.05.

## 3. Results

### 3.1. Temporal Parameters

The differences in temporal gait parameters measured in the non-pole (1), standard-pole (2), and mechatronic-pole (3) tests, taking into account the left and right sides of the measurement, are shown in Table 2. No differences were found for only three variables: cadence, step time, and stride time. For the other variables, as in the case of the ranges (Table 3), all the differences identified were between free-walking and walking with poles, with no differences between standard and mechatronic poles. Reduced values when a pole was used were found for double stance (by 26% for the regular pole only), load response (regular pole by 21% and 24% for LT and RT respectively, mechatronic pole by 23% and 16% for LT and RT respectively), pre-swing (regular pole by 23 and 21% for LT and RT respectively, mechatronic stick by 12% and 18% for LT and RT respectively), stance phase (by 4% for both sides and poles). Increases in values after the use of the standard and mechatronic pole relative to free-walking were observed for single support (standard pole by 6% and 4% for LT and RT, respectively, mechatronic pole by 2% and 4% for LT and RT, respectively) and swing phase (standard pole by 5% and 7% for LT and RT, mechatronic pole by 6% and 4% for LT and RT, respectively). For all temporal parameters, regardless of the use, or not, of the pole and its type, no asymmetry was confirmed between the right and left sides of the measurement (*p* > 0.05).

### 3.2. Kinematic Parameters

The results of all analyzed kinematic gait parameters across the three tests of without a pole (1), with a standard pole (2), and with a mechatronic pole (3) are shown in Table 3. For nine parameters, differences were found between free-walking and walking with poles for both left and right sides (1 vs. 2 and 1 vs. 3), while there were no differences due to the type of pole (2 vs. 3). 

Lower values during walking with poles compared to free-walking were observed for the ankle abduction–adduction range for the mechatronic pole (the relative percentage difference was almost 21% for the left and right sides) and ankle inversion–eversion for both poles (a reduction of 25% for the regular pole and 27% and 15% for the left and right sides, respectively, for the mechatronic pole).

Larger range values in walking with both poles than without were found for hip flexion–extension (regular pole up 8% and 12% for LT and RT, respectively, mechatronic pole up 10% and 15% for LT and RT, respectively), wrist radial–ulnar (regular pole up 77% and 110% for LT and RT, mechatronic pole up 92% and 98% for LT and RT, respectively), and wrist supination–pronation (regular pole up 74% and 21% for LT and RT, respectively, mechatronic pole up 76% and 52% for LT and RT, respectively). 

Differences between the measurements for the left and right sides were identified for only two parameters: shoulder abduction–adduction during regular pole walking and elbow flexion–extension in free-walking. For the other ranges, regardless of the use or not of poles and their types, there was no significant asymmetry between the right and left sides (*p* > 0.05).

## 4. Discussion

The purpose of the study was to compare selected temporal and kinematic parameters of normal gait and NW performed with classic and mechatronic poles (classic poles equipped with sensors). It was assumed that equipping NW poles with sensors for biomechanical gait analysis would not impair the NW walking technique. If the gait pattern of the group of professionals does not change, the prototype sticks have been correctly designed and can be used to monitor the correctness of the NW gait of other test groups.

### 4.1. Kinematic Parameters

Free-walking without poles over 10 m in an open space may differ from the gait recorded under laboratory conditions. This is because the kinematic variables of gait are influenced by the conditions under which the measurement takes place and, associated with this, the average speed obtained and the cadence (gait cadence) of the gait [22,23]. In the present study, for nine kinematic parameters, differences were found between free-walking and walking with poles for both the left and right sides, while there were no differences due to the pole type. For the upper limb, significant differences in ROM were observed for wrist radial–ulnar and wrist supination–pronation, while for the lower limb for ankle abduction–adduction, ankle inversion–eversion, and hip flexion–extension. The parameters for the lower limb that showed the greatest significant difference in ROM (*p* = 0.001) were the LT and RT hip flexion–extension. These parameters were significantly higher in walking with classic and mechatronic poles than walking without poles. This confirms the observations of some authors, who also found an increase in the flexion range at the hip joint during NW gait. However, it should be noted that there is a large variability in the results regarding the kinematics of the hip joint during NW gait, as some authors showed no difference in the range of motion at this joint in NW gait compared to free gait [3,7]. Increasing the range of hip flexion during NW certainly affects the lengthening of the swing phase and thus the shortening of the support phase during walking. 

Other kinematic parameters showed no difference between free-walking and NW, which may indicate that professionals use NW gait patterns in free-walking. In their case, the use of poles during walking did not affect their gait technique, while in those who were less advanced in walking with poles, it could be a disruptive variable for them, i.e., the poles could interfere with their walking to varying degrees.

Larger ranges of motion in the (radial–ulnar) wrist and ankle joints during NW compared to free-walking confirm the greater involvement of the upper and lower extremities resulting from the use of the poles. This is especially true for the ankle joint, where about 60% of the propulsive force is produced during gait. Increasing the range of motion in this joint during NW confirms its greater contribution and indicates that this type of gait increases this strength. This may be one of the reasons for the increased gait speed and improved economy in NW. 

The analysis that was conducted to assess the symmetry of the kinematic parameters showed asymmetry for only two: shoulder abduction–adduction in regular-pole walking and elbow flexion–extension in free-walking. 

### 4.2. Temporal Parameters

Analysis of temporal parameters showed statistically significant differences for the following parameters: time of stance phase, load response, single support, pre-swing, swing phase, and double stance. They show that during NW, the swagger and single-support phases were lengthened, while the double-stance phase was shortened. There were no statistical differences between groups for step time, stride time, and step frequency (cadence).

Other authors have obtained slightly different results, as they found an increase not only in single-support time, but also in double support during NW gait, by almost 10% and more than 20%, respectively, compared to free-walking (without poles) [3,5]. These differences are likely due to the study of groups with varying degrees of NW gait technique. In our study, these were NW athletes, coaches, and instructors, i.e., people who represented the highest level of NW gait technique. Other authors measured recreational NW walkers with little experience in this type of gait. The shortening of the double-stance phase is likely related to the increase in gait speed that characterizes the gait of professionals. This could have been another reason for the shortened double-support time during NW gait in our study. This would require additional studies comparing kinematics and gait kinetics depending on the level of walking technique. 

Moreover, the results of NW gait biomechanics studies available in the literature, which are relatively few in number and of varying quality, involve people with varying levels of activity and physical fitness and a large age range of 22 to 70 years. Analysis of NW gait biomechanics is also carried out in groups of patients with Parkinson’s disease, cardiovascular disease, osteoarthritis, or fibromyalgia. This variation in the study groups makes it impossible to reliably compare the results obtained by different authors [3]. 

It should be emphasized that the differences shown in our study apply only to free-walking and NW. No differences between the NW gait with a classic pole and a mechatronic pole were found. This confirms our assumption that equipping classic NW poles with additional signaling and measuring devices (mechatronic poles) would not affect movement technique, which allows their further use in the study of gait biomechanics. All temporal parameters, regardless of the use or not of the pole and its type, were characterized by the symmetry between the right and left sides of the measurement. 

The novelty of this study is the testing of utterly new NW poles, constructed by the Department of Fundamentals of Machine Design and Mechatronics Systems team at the Wroclaw University of Science and Technology, which had not yet been tested, used, or patented. The overarching aim of constructing these poles is to use them during rehabilitation by the NW method. Before practical use of the mechatronic poles in groups of patients with different clinical conditions, older adults, or people without NW experience, testing them on a professional group with extensive NW experience is necessary. Furthermore, the symmetry results in temporal and kinematic parameters (except for two) indicate perfect mastery of the NW walking technique by the group of professionals, allowing us to use the measurement results as a control in subsequent studies. 

### 4.3. Limitations of Research

While the study was carried out on people with the highest level of NW gait technique who are professionally involved in the sport (athletes, coaches, instructors), the relatively small number of subjects increases the risk of type 2 statistical error. The small sample size resulted from the limited number of volunteer NW instructors who agreed to participate in the study and the constraints due to the COVID-19 pandemic, during which the testing was conducted.

Furthermore, comparing the results obtained with those of other authors must be interpreted with caution due to the different levels of sophistication of the NW gait technique of the subjects studied. This is confirmed by the varied and inconclusive results of NW gait biomechanics studies in the available literature, which are likely due in part to the heterogeneous groups studied by different authors.

## 5. Conclusions and Applications

The use of both classical and mechatronic poles in gait increases the participation of the upper and lower limbs in the gait, causing an increase in the driving force of the gait, as well as its safety. Individuals representing the highest level of NW gait technique use the patterns of this gait in free gait. There were no statistically significant differences in the biomechanics of NW gait with classic and mechatronic poles. All spatial–temporal parameters were characterized by symmetry, while among kinematic parameters, only two were found to be asymmetric. Equipping classic NW poles with additional signaling and measuring devices (mechatronic poles) did not affect the movement technique, making it possible to use them in further studies of gait biomechanics.

## Figures and Tables

**Figure 1 ijerph-20-00163-f001:**
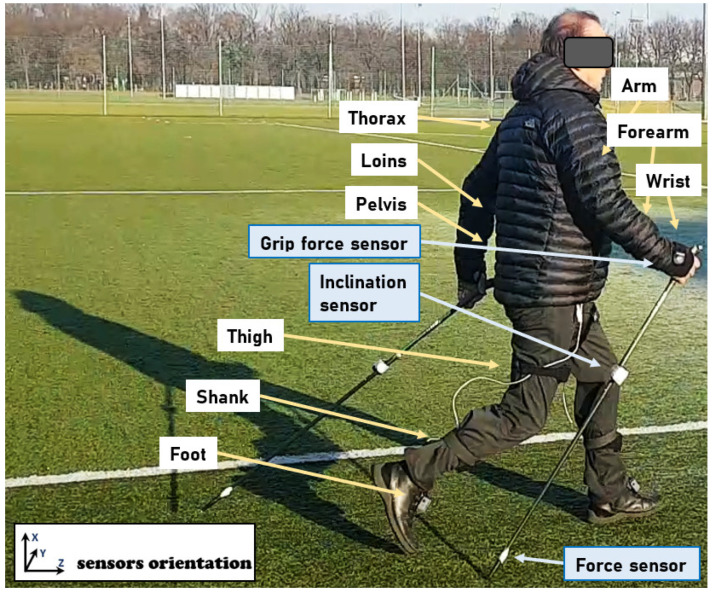
Illustration of sensors placement on the human body and the poles.

**Table 1 ijerph-20-00163-t001:** Characteristics of the participants.

Participants	Body Height (cm)	Body Mass (kg)	Seniority (Years)
Women			
Case1	178	57	10
Case2	169	58	10
Case3	165	55	15
Mean ± SD	170.66 ± 6.66	56.67 ± 1.53	11.67 ± 2.89
Men			
Case4	177	77	10
Case5	178	79	16
Case6	175	76	20
Mean ± SD	176.67 ± 1.53	77.33 ± 1.53	15.33 ± 5.03

Abbreviations: SD—standard deviation.

**Table 2 ijerph-20-00163-t002:** Differences between measurements of temporal gait parameters: without a pole (1), with a standard pole (2), and with a mechatronic pole (3) for Nordic Walking professionals (*n* = 6).

Parameters		Gait without a Pole (1) Mean ± SD	Gait with a Standard Pole (2) Mean ± SD	Gait with a Mechatronic Pole (3) Mean ± SD	*p*-Value for the ANOVA Test	Sign. Difference (1)–(2)–(3) (by Post Hoc Test)
Step cadence (step/min)		114.91 ± 8.16	114.56 ± 6.5	113.46 ± 8.04	0.726	
Stride time duration (s)		1.049 ± 0.077	1.052 ± 0.061	1.065 ± 0.075	0.603	
Step time duration (s)	LT	0.523 ± 0.038	0.519 ± 0.040	0.540 ± 0.053	0.652	
	RT	0.526 ± 0.040	0.532 ± 0.043	0.524 ± 0.026		
Stance phase (%)	LT	59.90 ± 1.16	57.71 ± 1.28	57.48 ± 1.13	0.001	(1)–(2)
	RT	59.8 0± 1.65	57.15 ± 3.01	58.18 ± 1.11		(1)–(3)
Swing phase (%)	LT	40.10 ± 1.16	42.29 ± 1.28	42.52 ± 1.13	0.001	(1)–(2)
	RT	40.20 ± 1.65	42.85 ± 3.01	41.82 ± 1.11		(1)–(3)
Single support (%)	LT	40.19 ± 1.6	42.43 ± 2.37	41.18 ± 2.16	0.004	(1)–(2)
	RT	40.06 ± 1.09	41.86 ± 1.50	41.78 ± 0.88		
Double support (%)		19.72 ± 2.65	14.58 ± 2.98	16.36 ± 1.76	0.019	(1)–(2)
Loading response (%)	LT	9.92 ± 1.07	7.81 ± 1.69	7.66 ± 0.73	<0.001	(1)–(2)
	RT	9.78 ± 1.74	7.48 ± 2.02	8.27 ± 1.11		(1)–(3)
Pre-swing (%)	LT	9.77 ± 1.80	7.51 ± 2.02	8.61 ± 1.57	0.001	(1)–(2)
	RT	9.95 ± 1.08	7.86 ± 1.68	8.15 ± 1.33		(1)–(3)

Abbreviations: SD, standard deviation; RT, right side; LT, left side.

**Table 3 ijerph-20-00163-t003:** Differences between measurements of kinematic gait parameters: without pole (1) vs. standard pole (2) vs. mechatronic pole (3) for Nordic Walking professionals (*n* = 6).

Parameters (deg)		Gait without a Pole (1) Mean ± SD	Gait with a Standard Pole (2) Mean ± SD	Gait with a Mechatronic Pole (3) Mean ± SD	*p*-Value for the ANOVA or *F Test*	Significant (1)–(2)–(3) (Post Hoc Test)
Upper limb movements						
Shoulder flexion–extension	LT	35.98 ± 9.08	35.8 ± 4.08	38 ± 8.61	0.662	
	RT	38.27 ± 10.68	38.4 ± 6.99	41.55 ± 10.33		
Shoulder ab/adduction	LT	12.23 ± 6.48	12.26 ± 3.83	15.48 ± 8.48	*0.262*	
	RT	13.69 ± 7.75	22.03 ± 8.24	21.36 ± 8.9		
Shoulder rotation	LT	31.56 ± 13.65	40.11 ± 12.49	38.31 ± 13.63	*0.549*	
	RT	48.15 ± 24	47.37 ± 23.86	44.12 ± 21.15		
Elbow flexion–extension	LT	40.26 ± 9.74	34.86 ± 5.41	32.09 ± 4.42	0.054	
	RT	31.2 ± 10.55	24.67 ± 12.32	25.77 ± 11.46		
Wrist flexion–extension	LT	16.92 ± 14.22	21.39 ± 5.58	20.95 ± 8.17	*0.532*	
	RT	14.1 ± 9.59	17.39 ± 4.4	15.93 ± 9.65		
Wrist radial–ulnar	LT	13.32 ± 6.24	23.58 ± 7.19	25.56 ± 3.06	0.001	(1)–(2)
	RT	10.65 ± 3.69	22.38 ± 5.09	22.16 ± 6.49		(1)–(3)
Wrist supination–pronation	LT	25 ± 12.26	43.53 ± 10.01	43.91 ± 8.14	0.022	(1)–(2)
	RT	39.37 ± 11.06	47.67 ± 15.1	59.95 ± 14.87		(1)–(3)
Lower limb movements						
Hip flexion –extension	LT	64.99 ± 7.29	70.42 ± 8.95	71.3 ± 6.76	0.001	(1)–(2)
	RT	63.04 ± 6.91	70.56 ± 7.04	72.17 ± 4.99		(1)–(3)
Hip ab/adduction	LT	16.5 ± 3.99	15.09 ± 3.26	14.63 ± 3.03	*0.563*	
	RT	16.79 ± 5.14	15.64 ± 1.92	16.9 ± 2.82		
Hip rotation	LT	21.1 ± 1.77	24.68 ± 3.52	24.65 ± 3.34	0.198	
	RT	24.3 ± 6.9	25.08 ± 7.7	26.12 ± 4.15		
Knee flexion–extension	LT	68.12 ± 2.67	67.92 ± 5.66	68.65 ± 2.24	*0.887*	
	RT	67.49 ± 3.91	66.68 ± 5.3	65.87 ± 4.74		
Ankle dorsi–plantarflexion	LT	35.5 ± 5.41	37.59 ± 10.09	36.52 ± 8.68	*0.698*	
	RT	40.38 ± 5.82	38.14 ± 10.12	35.16 ± 9.11		
Ankle ab/adduction	LT	18.3 ± 5.92	15.3 ± 4.87	14.5 ± 3.29	0.043	(1)–(3)
	RT	15.76 ± 4.63	14.46 ± 2.75	12.52 ± 2.31		
Ankle inversion–eversion	LT	19.22 ± 4.72	14.5 ± 2.96	14.07 ± 4.15	0.024	(1)–(2)
	RT	17.46 ± 5.53	13.17 ± 2.25	14.9 ± 6.11		(1)–(3)

Italics indicate Friedman test results; bold indicates significant differences between LT and RT unpaired-tests with Bonferroni correction (3 measurements); Abbreviations: SD, standard deviation; RT, right side; LT, left side.

## Data Availability

The data used to support the findings of this study are available from the corresponding author upon request.

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
