# Peer review of "No Influence of Mechatronic Poles on the Movement Pattern of Professional Nordic Walkers"

_ijerph, 2022, doi:10.3390/ijerph20010163_

Round 1
Reviewer 1 Report (New Reviewer)
Summary:
The purpose of this study was to assess kinematics of Nordic Walking experts between non-pole use, walking-poles use, and instrumented walking-poles. Six expert Nordic Walkers were assessed for upper and lower body kinematics using inertial measurement units (IMUs). Participants walked 100m in each condition. There were significant differences between the non-pole walking and pole-walking, but mot between pole-waling and instrumented pole-walking.
Comments:
The authors have provided a kinematics-based assessment of movement parameters in a small sample of expert Nordic Walkers. Although the data they report serve as a good baseline for reference, there are a number of issues that cannot be overlooked in this manuscript. Primarily, there are too many variables assessed based upon the number of participants (N = 6). Statistically, the p-values must be adjust based upon the number of tests which were performed. The authors mention in the Discussion a possibility of Type II error, but they did not provide a statistical correction for reducing this possible error. The following summary of each section highlights areas where the document can be improved.
Introduction: Overall, this section is well written and organized. The main question that is brought forward indicates that there is an assumption that instrumented waling poles would influence the Nordic Walking compared to normal Nordic Walking poles. However, it is not clear in the background information if this is a true issue in the assessment of Nordic Walking.
Methods: Information regarding the procedures performed and the equipment used are, in general, well presented. The list of dependent variables is extensive, and too many for this analysis to be effective. It is not clear what order the walking conditions were performed, and this should be mentioned. Participants walked 100m 6 times and it is not clear if there was a rest duration to reduce the influence of fatigue. Only three strides per condition were assessed, is this correct? This does not seem to be a sufficient number of trials to use considering that two trials of walking were performed per condition. For the gait parameters, it is not clear how foot contact was determined (change in the acceleration or an integration of the acceleration data to velocity). Similarly, how were stance and swing times established with the IMUs since no pressure sensors in the shoes were worn? It is unclear what ankle joint abduction/adduction really measures. The ankle joint can only move within the sagittal plane and is constrained in the frontal and transverse planes. Inversion/eversion is performed at the subtalar joint – but this is mentioned by the authors in the description of the movement variables. What is the difference between the ankle “ab/adduction” and subtalar inversion/eversion movements?
Statistical analysis – it is not clear what the authors were hoping to report with comparisons between right and left sides. Given the bilateral coordination of the task, there should be no significant discrepancy between the right and left, and this adds another variable to increase Type II error.
Results: The reporting of the results is sufficient, given the number of variables the authors analyzed. The number of analyses performed does raise concerns as to the actual significance of variables between the conditions. Was the gait velocity assessed between conditions?
Discussion: The structure of this section can be modified to re-state the purpose/hypothesis before assessing the main points of the results. The assessment of reduced foot rotations with the poles would indicate that the muscles of the lower extremities are not as engaged due to the increased base of support. In fact, reduced range of motion may provide a hinderance to the joint and muscles as they are not being actively engaged. The authors discuss the lack of research in this area and the difficulty in comparing their results to other studies. Again, the authors used a highly trained population, so this will be difficult when assessing their transferability to the general population.
Author Response
we wish to submit a revised manuscript titled “No influence of mechatronic poles on the movement pattern of professional Nordic Walkers” for your consideration.
Firstly, we would like to thank the Editor for allowing us the chance to revise the manuscript. We are also very grateful to the Reviewers who found the time to read and understand our work and gave us valuable comments that we had not noticed before. The comments are constructive for us and essentially strengthen the manuscript. We have tried our best to correct the alarming points in this revised manuscript according to the reviewers’ comments (all changes were tracked). The point-by-point responses are also provided in the attached file. Please let us know if the revised manuscript has something that needs to be additionally improved.
Furthermore, we also want to ensure that extensive language proofreading was carried out, which should improve the article’s clarity.
Sincerely yours,
Slawomir Winiarski
===================================================
1st Review’s Comments and Suggestions for Authors
Summary:
The purpose of this study was to assess kinematics of Nordic Walking experts between non-pole use, walking-poles use, and instrumented walking-poles. Six expert Nordic Walkers were assessed for upper and lower body kinematics using inertial measurement units (IMUs). Participants walked 100m in each condition. There were significant differences between the non-pole walking and pole-walking, but mot between pole-waling and instrumented pole-walking.
Comments:
The authors have provided a kinematics-based assessment of movement parameters in a small sample of expert Nordic Walkers. Although the data they report serve as a good baseline for reference, there are a number of issues that cannot be overlooked in this manuscript. Primarily, there are too many variables assessed based upon the number of participants (N = 6). Statistically, the p-values must be adjust based upon the number of tests which were performed. The authors mention in the Discussion a possibility of Type II error, but they did not provide a statistical correction for reducing this possible error. The following summary of each section highlights areas where the document can be improved.
- We are very grateful for the comment. We agreed with the suggestion and reduced the number of variables. Especially, data concerning trunk motion was removed as it was least relevant to our investigation. This concerns Table 3, Results and Discussion accordingly.
- In order to reduce the probability of Type I error, we used methods that take into account the adjusted p-value for the number of comparisons (multivariate ANOVA with Wilks’ correction and post-hoc tests performed only when the ANOVA confirmed the significance of the studied effect). We are aware of the likelihood of Type II error due to the small number of participants; therefore, we have stated this as a limitation of this study. We did not apply an adjustment to reduce the possibility of error because we did not have the possibility to increase the sample size. However, increasing the significance level even to α=0.10 does not imply (in most variables) the rejection of the null hypothesis. Therefore, to avoid increasing the risk of error of the first kind, the level of statistical significance was not adjusted.
Introduction: Overall, this section is well written and organized. The main question that is brought forward indicates that there is an assumption that instrumented waling poles would influence the Nordic Walking compared to normal Nordic Walking poles. However, it is not clear in the background information if this is a true issue in the assessment of Nordic Walking.
- The section was reorganized. We wanted to strengthen the argumentation and have provided a description of the frequently used sensors with instrumented poles. There is a widespread belief that the existence of such sensors significantly influences NW performance. In this article, we proved otherwise.
Methods: Information regarding the procedures performed and the equipment used are, in general, well presented. The list of dependent variables is extensive, and too many for this analysis to be effective.
- As we have mentioned earlier, following a discussion with the co-authors, we decided to remove the trunk variables (thorax, loins and pelvis; 3 DoF) - as the least relevant and least reproducible. The number of variables was substantially reduced.
It is not clear what order the walking conditions were performed, and this should be mentioned.
- In each case, they started with a free gait, and the order was random for the walk with poles; since these were professionals who had a learned movement technique. This is mentioned in L.153 and 154.
Participants walked 100m 6 times and it is not clear if there was a rest duration to reduce the influence of fatigue. Only three strides per condition were assessed, is this correct? This does not seem to be a sufficient number of trials to use considering that two trials of walking were performed per condition.
- The comment may be due to a misunderstanding. Approximately 70 strides (cycles) per passage were registered, totalling ~140 full strides (cycles) for each case. Short breaks were used between trials. For a group of professionals, a distance of 100 m was not tiring. We agreed to clarify our description and provided information about the rest period. It is included in lines 155 - 157.
For the gait parameters, it is not clear how foot contact was determined (change in the acceleration or an integration of the acceleration data to velocity). Similarly, how were stance and swing times established with the IMUs since no pressure sensors in the shoes were worn?
- The Noraxon’s myoMOTION system is equipped with virtual footswitches. These footswitches use the gyroscope and accelerometer data from the sensors assigned to the feet to determine foot contacts. The Virtual Footswitches allow users to see the steps of a subject visually and have the added benefit of allowing the user to utilise the MyoMotion Gait Foot Switch Report. The Report is based on the stance and swing phases to determine spatial gait parameters and display the averaged kinematic angle curves using the footswitch data to determine the time interval. The information was provided in L.204-206.
It is unclear what ankle joint abduction/adduction really measures. The ankle joint can only move within the sagittal plane and is constrained in the frontal and transverse planes. Inversion/eversion is performed at the subtalar joint – but this is mentioned by the authors in the description of the movement variables. What is the difference between the ankle “ab/adduction” and subtalar inversion/eversion movements?
- Yes, this is a common difficulty when comparing anatomical terms of movements to the definition of movements by the MAS systems - developed by engineers. According to the ROM description in Noraxon’s manual, ankle abduction is defined as a movement of the foot away from the centre line of the body, while adduction is a movement of the foot towards the centre line of the body. These movements take place with the heel stabilized. Whereas inversion refers to the pronation of the tarsus, and eversion refers to the supination of the tarsus. Nevertheless, we made sure our methodology was consistent with ISB standards. Mainly, Wu et al. “ISB recommendation on definitions of joint coordinate system of various joints for the reporting of human joint motion—part I: ankle, hip, and spine.” J Biomech. 2002;35:543–8.
Statistical analysis – it is not clear what the authors were hoping to report with comparisons between right and left sides. Given the bilateral coordination of the task, there should be no significant discrepancy between the right and left, and this adds another variable to increase Type II error.
- Gait is a cyclic, alternating movement, and in our experiment, all measurements were made on both sides, confirming good interlimb symmetry. This is a valuable result because the investigation is a part of a larger project; we have other studies on clinical groups in which significant asymmetries appear.
Results: The reporting of the results is sufficient, given the number of variables the authors analyzed. The number of analyses performed does raise concerns as to the actual significance of variables between the conditions. Was the gait velocity assessed between conditions?
- Regarding the first comment, we believe that the analyses we used indicate that the multiple comparisons reduced the likelihood of error of the first type. For each variable, an ANOVA with repeated measures was performed, including the effect of the R/L sides of the measurement. The post-hoc tests were performed only for the significant effects indicated by the ANOVA result (effect of poles used and/or the R/L side).
- As noted, the mean speed was not calculated; due to the lack of reliable spatial data. The speed was only controlled by measuring the walking time over the entire 100 m distance. For this reason, we also removed the name “spatial” from the parameters.
Discussion: The structure of this section can be modified to re-state the purpose/hypothesis before assessing the main points of the results.
- The structure of the Discussion has been completed (lines 273-279) by the following paragraph: “The purpose of the study was to compare selected temporal and kinematic parameters of normal gait and NW performed with the classic and mechatronic poles (classic poles equipped with sensors). It was assumed that equipping NW poles with sensors for biomechanical gait analysis would not impair the NW walking technique. If the gait pattern of the group of professionals does not change, the prototype sticks have been correctly designed and will be able to be used to monitor the correctness of the NW gait of other test groups.”
The assessment of reduced foot rotations with the poles would indicate that the muscles of the lower extremities are not as engaged due to the increased base of support. In fact, reduced range of motion may provide a hinderance to the joint and muscles as they are not being actively engaged. The authors discuss the lack of research in this area and the difficulty in comparing their results to other studies. Again, the authors used a highly trained population, so this will be difficult when assessing their transferability to the general population.
- Yes, it was our mistake that happened during the editing process. There is no such movement as ankle rotation (and never was such a result in our table). The controversial sentence “In contrast, the smaller ranges of rotational movement of the foot during NW gait may indicate the stabilizing role of the poles and confirm the greater safety of this type of gait and the lower risk of injury” WAS REMOVED. Thank you very much for this finding and the opportunity to eliminate this error.
We are, honestly, very grateful for all the valuable observations that we believe helped the article become a better academic text. We are also open to any other suggestions for additional improvements.
Reviewer 2 Report (New Reviewer)
Agnieszka Szpala et al. present an observational case series study aimed to compare selected spatial-temporal and kinematic parameters of normal gait and Nordic Walking (NW) performed with the classic and mechatronic poles (classic poles equipped with sensors).
- L-71 Such a system is able to identify the basic parameters that characterize correct movement technique and provide feedback on this to the user of the poles.
what kind of feedback (sound?), please explain in detail
- L-155 To obtain more information on the human gait information from gyroscopes and magneto resistive sensors, they are combined with accelerometers of the IMUs and applied in gait kinematics
Does the inertial sensor not already include the integration of an accelerometer, gyroscope and magnetometer? Please, explain the phrase or delete it
The analyzed sample of patients does not support the conclusions in the text, please replace chapter 5 with a discussion of the results and the feasibility study (or proof of concept study) on the use of sensors applied to the NW
Author Response
we wish to submit a revised manuscript titled “No influence of mechatronic poles on the movement pattern of professional Nordic Walkers” for your consideration.
Firstly, we would like to thank the Editor for allowing us the chance to revise the manuscript. We are also very grateful to the Reviewers who found the time to read and understand our work and gave us valuable comments that we had not noticed before. The comments are constructive for us and essentially strengthen the manuscript. We have tried our best to correct the alarming points in this revised manuscript according to the reviewers’ comments (all changes were tracked). The point-by-point responses are also provided in the attached file. Please let us know if the revised manuscript has something that needs to be additionally improved.
Furthermore, we also want to ensure that extensive language proofreading was carried out, which should improve the article’s clarity.
Sincerely yours,
Slawomir Winiarski
2nd Reviewer’s Comments and Suggestions for Authors
Agnieszka Szpala et al. present an observational case series study aimed to compare selected spatial-temporal and kinematic parameters of normal gait and Nordic Walking (NW) performed with the classic and mechatronic poles (classic poles equipped with sensors).
- L-71 “Such a system is able to identify the basic parameters that characterize correct movement technique and provide feedback on this to the user of the poles.” what kind of feedback (sound?), please explain in detail.
- The sentence in question – description of a design work of mechatronic poles equipped with sensors - was changed entirely. Generally, audible or visual signals are used, but this was not applied in our poles until later. At this stage of the project, a recording of gait parameters on a tablet was used; which could be analysed, but offline rather than in real-time. The changed description is in L.70-81: “Work is underway on the design of mechatronic poles equipped with sensors that would allow real-time analysis of these parameters and, based on this, correction of the gait technique. Such a system can identify the basic parameters that characterise correct movement technique and provide feedback on this to the user of the poles. One such system component is a wireless communication module that transmits the measured NW gait parameters to a mobile device (smartphone/tablet). A dedicated application is installed on the mobile device, which enables online/offline monitoring and analysis of NW gait by the trainer, physiotherapist or patient [13]. Among other things, the app allows the visualisation of selected waveforms in the form of dynamic graphs and displays animations of pole movement. The application includes a module for defining events, the occurrence of which is signalled by sound/vibration (e.g. signalling the moment the pole contacts the ground or is released); it can define the angular range of the movement and signal its exceedance, etc.) [13]”
- L-155 “To obtain more information on the human gait information from gyroscopes and magneto resistive sensors, they are combined with accelerometers of the IMUs and applied in gait kinematics”. Does the inertial sensor not already include the integration of an accelerometer, gyroscope and magnetometer? Please, explain the phrase or delete it.
- Thank you for this observation. This sentence has also been reformulated. It now reads: “To obtain more information on the human locomotion, additional IMU sensors were used to register the movements of human body parts.”
The analyzed sample of patients does not support the conclusions in the text, please replace chapter 5 with a discussion of the results and the feasibility study (or proof of concept study) on the use of sensors applied to the NW.
- Also, in this case, we agree with the comment. The bulleted conclusions were turned into a concise paragraph on Final remarks and applications. Now in lines 369-378.
We are, honestly, very grateful for all the valuable observations that we believe helped the article become a better academic text. We are also open to any other suggestions for additional improvements.
This manuscript is a resubmission of an earlier submission. The following is a list of the peer review reports and author responses from that submission.
Round 1
Reviewer 1 Report
The article examines the influence of using instrumented versus uninstrumented poles in Nordic walking.
The article analyzes spatiotemporal parameters and angular kinematics in 6 athletes during walking without poles, with regular poles, and with instrumented poles.
The authors conclude that the use of instrumented poles does not influence any walking parameters.
In my opinion, the work, while generally well structured, does not have a degree of novelty, nor the to justify its publication in a journal. In many other studies, especially on cross-country skiing, instrumented poles have already been used instead of regular ones and no differences have been reported. In addition, the number of subjects is small (6 subjects), all of whom are professionals. This implies that no conclusions can be drawn from such a small sample.
If the authors' concern is the invasiveness of the system they present compared to other wireless solutions already in the literature, the article is not interesting for the readers because it tests a solution that is not state of the art. The subject population also turns out to be peculiar: professionals trained to repeat the gesture over and over again, no matter the poles they are using.
Here after some minor comments:
in table 1: the mean age of the participant is missing
the description of the athletes lines 99-110 allows to recognize the subjects
lines 118 – 128 since also the magnetic unit ids present and used in the Myomotion MR3, it is correct to use MIMU and not IMU. Please correct in all the paper
lines 134 – 146: since one of the paper authors refer to describe the instrumented pole is in Polish, I suggest to add a brief description of the instrumented poles, including the total weight. In Figure 1 would be advisable to use the instrumented poles instead of normal poles.
Reviewer 2 Report
Thank you for the opportunity to read this manuscript. This study is very interesting and the results could be clinically relevant for clinicians to instruct walking exercises. However, there are a few concerns. As the authors mentioned in the limitation section, the sample size was too small (n=6) even they were highly experienced level. Are the data normality distributed? There may be a type 2 error. In Table 3, the significant was indicated with bold characters (I assume). However, the p-value was not significant. Further, there is a lack of IMU information. Is the sensor original? Please add information of company, reliability and validity data (not just high). I highly recommend the authors to get more sample data and reanalyze.
Reviewer 3 Report
The purpose of this study is to compare the gait parameters among NW gait, NW gait with pole and with mechatronic pole.
The paper is very well written, the content is interesting and relevant. I just have a few comments to improve the comprehension of the reader:
- introduction: distance and gait speed would be better classified as spatial temporal parameters, instead of physiological;
- methods: in my opinion, there is need for the explanation of the definition of the spatial temporal parameters. They are known in the biomechanical community, so I think it could be removed.
- statistics: it is not clear that you have used the 70 gait cycles for each subject individually, I mean, it is not the average of the 70 for each person. It seems that this was done because the distribution was normal, but could be more clear.
- Results: Table 2 - the variables are percent of what? again, n=6 is not entirely true, since it was used many cycles for each person, which can alter the results obtained.
- discussion: I think that a paragraph in the beginning of the discussion, reminding about the purpose of the study and the summary of the methods used helps the the reading.
Also, since the results were presented first spatial temporal and then kinematics, the discussion should follow the same sequence.
Please insert the applicability of the results in future investigations and in the improvement of NW technique.
Conclusion: the conclusion is just a summary of the main results, and it should be a highlight about the implications of these results for the NW research..